# Overview of Immune Checkpoint Inhibitors in Gynecological Cancer Treatment

**DOI:** 10.3390/cancers14030631

**Published:** 2022-01-27

**Authors:** Boštjan Pirš, Erik Škof, Vladimir Smrkolj, Špela Smrkolj

**Affiliations:** 1Faculty of Medicine, University of Ljubljana, 1000 Ljubljana, Slovenia; bostjan.pirs@kclj.si (B.P.); eskof@onko-i.si (E.Š.); vladimir.smrkolj@student.mf.uni-lj.si (V.S.); 2Division of Gynaecology and Obstetrics, University Medical Centre, 1000 Ljubljana, Slovenia; 3Department of Medical Oncology, Institute of Oncology Ljubljana, 1000 Ljubljana, Slovenia

**Keywords:** immune check-point inhibitor, biomarker, endometrial cancer, ovarian cancer, uterine cervical cancer, vulvar cancer, treatment response prediction

## Abstract

**Simple Summary:**

Recently, cancer treatment has been revolutionized by introduction of immunotherapy—drugs that target body’s immune system to attack cancer. Most clinically used drugs stop the mechanisms that dampen immune response. These drugs are called immune checkpoint inhibitors (ICIs). ICIs in gynecological cancers are most effective for treating uterine endometrial cancer, but less so far ovarian, uterine cervical or vulvar cancer. However, combining ICIs with other drugs has yielded good results in some studies in these cancers. Stopping mechanisms that dampen immune response can produce severe side effects, as has been seen with the use of ICIs. Therefore, selection of patients that would benefit the most from ICI therapy is of paramount importance. This can be done by analysing tumour characteristics either by looking at protein expression, genetic changes and even constitution of faecal microbiota, these properties are called biomarkers. It is not entirely known which biomarkers predict response most accurately, and this varies by cancer type. In this article, we review mechanisms of action of ICIs, selected biomarkers and latest clinical trials of ICIs in gynecological cancers.

**Abstract:**

In the last ten years, clinical oncology has been revolutionized by the introduction of oncological immunotherapy, mainly in the form of immune checkpoint inhibitors (ICIs) that transformed the standard of care of several advanced solid malignancies. Using ICIs for advanced gynecological cancers has yielded good results, especially for endometrial cancer. In ovarian or cervical cancer, combining ICIs with other established agents has shown some promise. Concurrently with the clinical development of ICIs, biomarkers that predict responses to such therapy have been discovered and used in clinical trials. The translation of these biomarkers to clinical practice was somewhat hampered by lacking assay standardization and non-comprehensive reporting of biomarker status in trials often performed on a small number of gynecological cancer patients. We can expect increased use of ICIs combined with other agents in gynecological cancer in the near future. This will create a need for reliable response prediction tools, which we believe will be based on biomarker, clinical, and tumor characteristics. In this article, we review the basic biology of ICIs and response prediction biomarkers, as well as the latest clinical trials that focus on subgroup effectiveness based on biomarker status in gynecological cancer patients.

## 1. Introduction

The idea of activating the immune system against cancer dates back as early as the 19th century, with experiments by William Coley, who injected live or inactivated pathogens into tumors. However, until recently, contemporary oncological practices did not take advantage of this mechanism, at least not directly. In the last ten years, the field has been revolutionized by the introduction of oncological immunotherapy, most notably by the development of a new class of systemic biological therapy directed towards immune receptors and their ligands, so-called immune check-point inhibitors (ICIs) [1]. These agents transformed the standard of care of several solid tumors, including classically difficult-to-treat tumors, such as metastatic melanoma and non-small cell lung, urothelial, and kidney cancer [2]. Based on the mechanism of action of these agents, several biomarkers for the response to treatment have been tested in clinical trials, which have led to regulatory approvals of ICIs based on the presence of these biomarkers. Furthermore, this has led to tissue-agnostic approvals in which an anticancer drug is approved not according to its histology but solely on the presence of a biomarker. More recently, trials of ICIs for gynecological cancer have produced promising results, especially for endometrial and partially for uterine cervical carcinoma. Gynecological cancers represent a heterogenous group of tumors, and thus their responses to ICIs can be predicted by using several biomarkers. However, the optimal biomarkers for a specific type of cancer have not yet been fully determined [1,2,3,4].

In this article, we review the basic biological mechanisms of ICI action and the biological bases of biomarker selection. We provide a state-of-the-art review of the clinical trials of ICIs utilized in gynecological cancers and summarize current USA- and Europe-based clinical guidelines on this topic.

## 2. Methodology

For this narrative review, we searched the PubMed database with the following search phrase: “(“Ovarian Neoplasms”[Mesh] OR “Endometrial Neoplasms”[Mesh] OR “Uterine Cervical Neoplasms”[Mesh] OR “Vulvar Neoplasms”[Mesh]) AND “Immune Checkpoint Inhibitors”[Mesh])”. Results were limited to the ones published in the last 5 years. Unrelated studies were excluded through careful browsing of the title, abstract, and/or whole text of each publication. Literature not limited to gynecological cancers was discovered with the method of snowballing. This review contains information and data from 139 papers, narrowed down from 214 relevant articles.

## 3. Mechanism of Action and Development of ICIs

Mechanism of action and development of ICIs is shown in Figure 1.

### 3.1. Overview of Tumor Immunobiology

The steps in the immune response to tumors are described below. Cancer-related antigens are novel proteins that result from mutations that cancer cells harbor (so-called neoantigens) or normal proteins that are strongly expressed in cancer cells (so-called cancer antigens). Dendritic cells in the vicinity take up these antigens but require activation by secondary signals to initiate the immune response. Such signals can be present in the tumor microenvironment (TME). In lymph nodes and other lymphatic tissue, activated dendritic cells present these antigens to effector T cells, establishing specific cell immunity. In the absence of activation signals for dendritic cells, regulatory T cells proliferate, promoting immune tolerance. Furthermore, inhibitory complexes of proteins on T cells and dendritic cells can stop effector T-cell activation and instead promote regulatory T-cell expansion, e.g., the protein CTLA4 on T cells. Effector T cells (CD8+) perform their function only after entering the TME, after which they are called tumor-infiltrating lymphocytes (TILs). Larger numbers of TILs in the TME correlate with good prognosis, whereas larger numbers of regulatory T cells in the TME correlate with poor prognosis.

Cancer cells employ several mechanisms to escape immune attacks. For example, they can downregulate MHC I class protein expression and upregulate inhibitory proteins that form complexes with respective receptors on T cells. This downregulation can occur via genetic mechanisms, e.g., a mutation in antigen-presenting protein beta-2-microglobulin (B2M); or epigenetic mechanisms. Examples of inhibitory proteins are PD-L1 and PD-L2, which form complexes with PD-1 on T cells, resulting in T-cell deactivation, termed anergy or T-cell exhaustion. Exhaustion occurs in states of chronic inflammation with chronic antigen exposure and persistent T-cell activation. After some time, the effector functions of CD8+ T cells diminish, for which a marker is a stronger expression of inhibitory receptors and ligands on their surface. Chronic autoimmune diseases and TMEs are examples of such states. TME is immunologically modulated by cytokine pattern and immune cell infiltration pattern, including pro-tumoral macrophages and myeloid-derived suppressor cells. Hence, PD-L1 or PD-L2 expression on tumor cells and high PD-1 expression on TILs correlates with poor prognosis. Additional mechanisms of cancer cells escaping immune attack include the release of various small molecules, such as indoleamine dioxygenase or adenosine (the production of which is induced by hypoxia, typical for the TME). The migration of T cells to the TME can be hampered by vascular cells, which function differently in tumors compared to normal tissue. This effect is partly mediated by vascular endothelial growth factors [1,3,4,5,6,7,8,9].

### 3.2. Overview of Immune Checkpoint Inhibitors

The co-inhibitory receptors and ligands mentioned above serve as targets for the development of their respective inhibitors—ICIs. The ICIs currently in clinical practice target the following: CTLA-4 (ipilimumab and zalifrelimab), PD-1 (nivolumab, pembrolizumab, dostarlimab, cemiplimab, and balstilimab), and PD-L1 (avelumab, durvalumab, and atezolizumab). All these agents are monoclonal antibodies that are active when administered parenterally. Furthermore, agents that are active orally or target other immune inhibition and activation pathways are currently being developed [2,10,11].

The rationale for combining ICIs with chemotherapy and other antineoplastic agents is based on the premise that they make the TME more immunoreactive. Some hypothesized and proven mechanisms are described here. Radiotherapy or chemotherapy results in DNA damage and subsequently immunogenic cell death with T-cell activation. Agents that prevent DNA damage repair (e.g., poly adenosine diphosphate-ribose polymerase (PARP) inhibitors) can lead to a higher mutational burden, subsequent higher neoantigen load, and activation of cytokine expression in tumor cells by releasing damaged DNA into the cytosol. Furthermore, antiangiogenic agents normalize tumor vasculature, which enhances immune cell migration into the TME. Tyrosine kinase inhibitors inhibit signaling pathways that mediate the conversion of tumors to cold tumors through several mechanisms that are not yet fully elucidated [12,13,14,15,16].

### 3.3. Immune Checkpoint Inhibitors—Related Adverse Events

The profile of adverse events (AEs) of ICIs differs from that of chemotherapy or targeted therapy. AEs caused by activation of the patient’s immune system, termed immune-related AEs, are most commonly transient to moderate. Common AEs include fatigue, mild infusion reactions, inflammatory skin reactions, colitis with diarrhea, hepatotoxicity, and mild endocrinopathies (e.g., hyper- or hypo-thyroidism). Rare AEs include pneumonitis, severe colitis, severe endocrinopathies (e.g., hypophisitis), adrenal insufficiency, type 1 diabetes mellitus, and rheumatological, hematological, and neurological symptoms (e.g., headache or peripheral sensory neuropathy). Severe and potentially fatal yet rare AEs include cytokine release syndrome, acute kidney injury, Guillain–Barre syndrome, myocarditis, and severe forms of the otherwise common AEs described above. AEs are more common after the use of anti-CTLA4 vs. anti-PD-1/PD-L1 ICIs. This is attributable to a mechanism of action. Anti-CTLA4 ICIs work more upstream in immune activation sequence-preventing CTLA4 to impede the acquisition of T cell effector function. On the other hand, anti-PD-1/L1 ICIs work downstream, preventing suppression of already differentiated effector T cells in peripheral tissues. Combining both classes of ICIs dramatically increases the incidence and severity of AEs, i.e., the rates of grade 3 and 4 AEs by 55%. Interestingly, the tumor site correlates with the type and grade of AEs, e.g., pneumonitis is more common when using ICIs to treat lung cancer. Additionally, immune-related AEs have been shown to predict treatment response to ICIs to some extent [2,17,18,19].

## 4. Biomarkers That Predict the Response to ICIs

Selected ICI treatment response prediction biomarkers are summarised in Table 1.

### 4.1. TILs

The presence of TILs in the TME correlates with the response to ICIs. According to the extent and pattern of infiltration, tumors can be classified as either (1) immune-inflamed (i.e., hot tumors, with TILs present between tumor cells), (2) immune-excluded (with TILs present in the stroma but not in the nests of tumor cells), or (3) immune-deserted (with a total absence of TILs in the TME). The latter two groups are termed non-immunoreactive with, unsurprisingly, a poor response to ICI treatment. The extent of TME infiltration can be assessed semi-quantitatively using hematoxylin and eosin staining or immunohistochemistry (IHC) that targets CD8+ cells. To date, no regulatory approval of ICIs has been based upon the presence of TILs in tumor specimens [2,14,20,21].

### 4.2. PD-L1 Expression

In line with the principles of tumor immunobiology outlined in Section 3.1, PD-L1 expression in tumor tissue was demonstrated to correlate with ICI treatment responses in various solid tumors in several studies [22]. All assays for measuring expression are semi-quantitative and use IHC on tumor tissue. However, different assays use different antibodies and platforms, which may not yield congruent results. Additionally, positive PD-L1 expression is not standardized, as one can measure expression on tumor cells, i.e., the tumor proportion score (TPS), and in the entire TME including immune cells, i.e., the combined proportion score (CPS), previously known as the modified H score or modified proportion score (MHS/MPS). Furthermore, the spatial pattern of IHC staining for PD-L1 (namely, at the tumor–stroma border, i.e., the interface pattern) can be included in the assessment. In clinical trials of ICIs, various cut-offs for TPS or CPS were used to stratify patients, varying from 1% to 50%. Nevertheless, the first regulatory approval of ICIs on the basis of PD-L1 expression was established for non-small cell lung cancer [22,23]. In gynecological cancers, PD-L1 was found to be a useful biomarker in cervical and, to some extent, in epithelial ovarian carcinoma, as described below.

### 4.3. TME Gene Expression Profiles

Initiation of the immune response in the TME leads to upregulated expression of several immune molecules by immune cells. Examples are cytokines (e.g., interferon-gamma), intracellular signaling molecules, chemokines, and molecules that inhibit further immune activation (e.g., the checkpoint molecules PD-L1 and PD-L2). Ayers et al. evaluated gene expression profiles by quantifying RNA from tumor specimens and correlated the profiles with the response to ICI therapy, thus producing the score from a predictive model based on the expression of 18 genes. The score can be considered to represent the degree of the immune response in the tumor, a marker similar to TIL number or PD-L1 expression. This scoring system was used in one ICI study; however, no regulatory approval based on this marker has been established yet [24,25].

### 4.4. Tumor Neoantigen Load and Mutational Burden

The hypothesis that the more mutations a tumor has, the higher the number of neoantigens that the tumor cells will produce (and hence the more robust the immune response) has been, at least partially, confirmed. Aside from the total number of neoantigens, a specific neoantigen expression pattern is required to produce an immune response [26,27,28]. Tumor neoantigen load is difficult to determine in vitro but can be predicted by in silico analysis of tumor DNA obtained by next-generation sequencing (NGS) [29,30]. Therefore, a surrogate marker that assesses global mutation rates in tumor cell genomes was proposed as a biomarker for the response to ICI therapy, i.e., tumor mutational burden (TMB). The gold standard of assessing TMB is NGS of coding regions, i.e., whole-exome sequencing. However, simpler and cheaper methods that target limited regions of the exome (cancer-related genes) have been proposed and validated recently, notably the FoundationOne and MSK-IMPACT assays [26,31,32,33]. TMB has been correlated with the response to ICI therapy [34]. The definition of TMB-high (TMB-H) remains to be established, and the thresholds depend on the assay used, e.g., for the FoundationOne assay, values between 9.9 and 20 mutations per million base pairs (mega base pair; Mb) were proposed [26,31,34,35,36,37]. Considering a threshold of 10 mutations/Mb, the prevalence of the TMB-H status in solid tumors averages approximately 13%, varying from 5–10% in certain esophageal and breast carcinoma subtypes to as high as 50% in melanoma. Recent research discovered that not all TMB-H tumors respond favorably to ICI treatment; however, cancers in which TILs levels correlate with neoantigen load do respond favorably. As for gynecological cancers, TMB-H predicts good responses in endometrial and cervical but not in ovarian carcinomas [38]. Pembrolizumab has been tissue-agnostically approved for the treatment of metastatic cancers in patients with no satisfactory treatment left based on TMB-H status. Thus, TMB may be potentially used for gynecological cancers as well.

### 4.5. Microsatellite Instability (MSI) and Mismatch Repair (MMR) Deficiency

Tumors harboring a defective mismatch repair (dMMR) mechanism have a defect in at least one of the four proteins involved in this specific DNA repair mechanism (MLH1, MSH2, MSH6, and PMS2). MMR deficiency is detected by IHC using antibodies against the four proteins mentioned above. These proteins function as heterodimers (MLH1 partners with PMS2, and MSH2 partners with MSH6), and inactivation can occur due to germline mutations (Lynch syndrome), somatic mutations, or epigenetic silencing (gene promotor methylation). According to the results, tumors are categorized as MMR-deficient (dMMR) or MMR-proficient (MMRp). IHC of dMMR tumors has its disadvantages, as non-conclusive results can be obtained, and other tests of dMMR are needed (e.g., MSI status). Microsatellites, synonymous with short tandem repeats, are repetitive DNA sequences dispersed throughout the genome. The numbers of these repeats are polymorphic in the population but constant in an individual. Because of their repetitive nature, they are prone to alterations in the number of repeats due to dMMR mechanisms. MSI testing is performed by PCR on DNA isolated from tumor tissue. Five probes are used, and if the results for at least two probes show altered repeat lengths, then MSI is confirmed as high (MSI-H). Conversely, tumors that do not show MSI-H are termed microsatellite-stable (MSS). Without an emphasis on the type of test used, MSI-H/dMMR and MSS/MMRp abbreviations are commonly used in studies.

Another technique for testing MSI employs NGS [35,39,40]. Deficiency in DNA repair mechanisms results in a hypermutated genome, rendering dMMR or MSI a surrogate marker of tumor mutational burden. There is some overlap between MSI-H and TMB-H, e.g., in an analysis of 62,000 tumor specimens, 5.8% were only TMB-H, 0.2% were only MSI-H, and 1.1% were both [26]. In general, TMB-H is more common, whereas MSI-H is less common. Additionally, most MSI-H cases are TMB-H as well, whereas most TMB-H cases are not MSI-H. The overlap and prevalence vary with the cancer type, with high concordance in cancers that are often MSI-H, e.g., endometrial and colorectal carcinomas [26,35]. Hence, other mechanisms, in addition to MSI, confer high TMB, e.g., the polymerase epsilon (POLE) mutation in endometrial cancer [35,39,41]. MSI/MMRp is an important biomarker for ICI responses in endometrial cancer, and ICIs have already been approved based on MMR or MSI status.

### 4.6. Microbiome

Recent studies indicate that gut microbiota influences the efficacy of PD-L1 and CTLA4 directed immune checkpoint inhibition therapies. For example, antibiotics administration lowered ICI efficacy in several studies. Furthermore, several bacterial strains associated with better response were identified, notably Akkermansia mucinophila, Enteroccocus hirae, Bifidobacterium spp. Responses for some strains were inconsistent, e.g., Bacterioides spp. Mechanism of gut microbiota influencing cancer immune response is not yet fully elucidated, but it has been shown it is associated with differential expression of cytokines, notably interferon-gamma, interleukin −12 and −17. Microbiome properties are not yet routinely clinically used to predict ICI treatment response. However, the findings described above are being used to enhance ICI response by means of fecal transplantation, as has been demonstrated in two recent clinical trials [42,43,44,45].

## 5. ICIs in Gynecological Cancer

### 5.1. Endometrial Cancer

Representing the most common gynecological cancer in the developed world, endometrial cancer has an incidence and prevalence that is still rising, with an estimated 121,000 new cases and 30,000 deaths in Europe in 2018 [46]. Although patients diagnosed at an early stage have a good 5-year survival of 95%, patients diagnosed at a late stage have a dismal prognosis with a 5-year survival of only 17%. Therapies for patients with recurrent or metastatic disease were until recently limited, usually consisting of platinum-based chemotherapy and hormonal therapy. In the standard-of-care chemotherapy regimen used in first-line settings (carboplatin plus paclitaxel), the median progression-free survival (PFS) and overall survival (OS) were 13 and 37 months, respectively. Before the advent of immunotherapy and targeted therapy, the options for patients with progression after first-line systemic chemotherapy were retreatment with carboplatin plus paclitaxel, single-agent chemotherapy, or hormonal therapy; the median OS was generally <12 months [46,47,48,49]. 

The Cancer Genome Atlas classified endometrial cancer into four molecular subtypes: (1) DNA POLE-mutated, (2) dMMR/MSI-H, (3) copy number high, and (4) copy number low. A technically more easily obtainable classification based on the p53 mutation instead of copy number variation was subsequently developed and incorporated into management guidelines [50,51,52]. Interestingly, the POLE mutation correlates with high-grade histology but has a good prognosis. The proposed explanation is that these tumors trigger robust immune responses. It is postulated that they are ideal candidates for immunotherapy, which is seldom indicated because first-line therapies (surgery) often suffice [53].

As endometrial cancer has solid tumors with the greatest incidence of MSI-H (30% of cases), it is expected that ICIs should be effective. Interestingly, the prevalence of TMB-H tumors is approximately 20%, and although approximately 30% of cases overlap, a large proportion of tumors are MSI-H without TMB-H, and a smaller proportion of tumors are TMB-H without MSI-H [31,35,39]. Specific to endometrial cancer, the POLE-mutated phenotype exhibits an extremely high mutation rate with an average of 232 mutations/Mb. dMMR/MSI-H and MSS tumors exhibit average mutation rates of 18 and 3 mutations/Mb, respectively. POLE-mutated tumors are highly correlated with PD-L1 expression and TIL presence in the TME (with a prevalence of 83%). MSI-H tumors have a somewhat lower but still high prevalence of TILs (78%) and PD-L1 expression (56%) [50,53]. PD-L1 is, however, often expressed in endometrial carcinoma tumors, including MSS/MMRp [35]. Most clinical trials of ICIs in endometrial cancer focused on MSI/MMR status, which turned out to be an excellent biomarker of the immune response as described below.

Studies of ICIs in MSI-H/dMMR endometrial cancer (EC) have shown promising results. In KEYNOTE-158, a phase 2 trial, pembrolizumab was administered to 49 MSI-H patients with metastatic/recurrent disease, of whom the majority had received several lines of systemic therapy beforehand. The overall response rate (ORR) was 57%, the median PFS was 25.7 months, and 16.3% of patients had a complete response [37]. GARNET was a phase 1 trial of dostarlimab with 103 dMMR patients with metastatic/recurrent disease who progressed during or after platinum-based chemotherapy, and the cohort A1 had an ORR of 46%, a disease control rate (DCR) of 59%, and 10.7% of patients with a complete response [54]. 

Responses to ICIs MSS/MMRp EC were worse than those in MSI-H /dMMR EC. MSS/MMRp in the GARNET trial (142 patients) had an ORR of 13% and a DCR of 35% [54]. In the PHAEDRA trial, a phase 2 trial (35 patients) of durvalumab, the MMRp cohort had an ORR of only 3% (one partial response) and a DCR of 19% [55]. In a phase 2 trial of avelumab (16 patients), only one objective response was observed, i.e., the ORR was 6.25% [56]. 

In both the KEYNOTE-158 and GARNET trials, a TMB marker predicted a better response to ICI even in the MSS/MMRp cohort. Specifically, in the KEYNOTE-158 trial, the ORR was 46% in the TMB-H subgroup vs. 6.0% in the non-TMB-H subgroup. In the GARNET trial, the ORR in TMB-H MSS/MMRp patients was 45.5% vs. 12.1% in TMB-low patients. Of note, there were no patients with POLE-mutated endometrial cancer in the TMB-H group in the GARNET trial. TMB-H was defined as at least 10 mutations/Mb [37,54,57,58]. The results indicate the potential of TMB as an additional biomarker of the ICI response in EC patients.

Recently, combinational therapies have been studied, mostly in MSS/MMRp patients. The rationale is that the addition of chemotherapy or other antineoplastic agents modulates the tumor’s immune environment, making ICIs more effective [59]. KEYNOTE-775 was a phase 3 trial on 827 patients (of those, 643 MMRp patients). Combining the ICI pembrolizumab with the antiangiogenic agent lenvatinib resulted in a median PFS and OS of 6.6 and 17.4 months, respectively, compared to 3.8 and 12 months, respectively, after single-agent chemotherapy [60]. Another interesting phase 2 trial combined nivolumab and cabozantinib, a multi-tyrosine kinase inhibitor that blocks the vascular endothelial growth factor receptor as well. The trial included 82 patients with at least one prior platinum-based chemotherapy and randomized them to either nivolumab alone or combination therapy. The ORR and median PFS was 25% and 5.3 months, respectively, in the combination group, compared to 11.1% and 1.9 months, respectively, in the nivolumab group [61].

Several ongoing trials are studying the combination of chemotherapy with ICI. These include ATTEND (atezolizumab and platinum-based doublet), NRG-GY018 (pembrolizumab with platinum-based doublet), NSGO-RUBY (dostarlimab with platinum-based doublet and subsequently the PARP inhibitor niraparib), and DUO-E (durvalumab with the PARP inhibitor olaparib).

Based on the results of the studies above, several ICIs were approved for use in endometrial cancer. Namely, pembrolizumab was FDA-approved in the USA in 2017 for tissue-agnostic treatment of MSI-H/dMMR or TMB-H tumors with progression following prior treatment and no satisfactory alternative options. Dostarlimab was approved in the USA and EU in April of 2021 for the treatment of MSI-H/dMMR dMMR/MSI-H advanced or recurrent endometrial cancer.[36,62,63] Pembrolizumab in combination with lenvatinib was approved for the treatment of advanced MSS/MMRp EC patients in the USA in July of 2021 and in Europe in November of 2021 [64,65].

As of November 2021, the European Society of Medical Oncology (ESMO) guidelines mention consideration of ICIs for MSI-H/dMMR endometrial cancer in second-line advanced/recurrent settings with a note that dostarlimab is approved in the EU [66]. The USA-based National Comprehensive Cancer Network (NCCN) guidelines recommend MSI/MMR and TMB testing for recurrent endometrial cancer, pembrolizumab for MSI-H/dMMR, or TMB-H tumors, and the combination of lenvatinib with pembrolizumab for MSS/MMRp tumors [67].

### 5.2. Uterine Cervical Cancer

Uterine cervical cancer is the fourth most common cancer in females, with 604,000 new cases and 342,000 deaths worldwide in 2020. As its incidence and mortality depend on the presence and quality of screening programs, 85% of cases occur in resource-limited regions [68,69]. In lower stages, the disease has a good prognosis, which increasingly worsens with higher stages, e.g., the 5-year survival is 64% in stage IIB (tumor involves parametria), 61% in stage IIIC1 (positive pelvic lymph nodes), 38% in stage IIIB (positive para-aortic lymph nodes), and 15% in stage IVB (parenchymal organ metastases) [70]. First-line treatment for metastatic or advanced cervical cancer not amenable to surgical resection or radiotherapy consists of platinum-based chemotherapy in combination with the antiangiogenic agent bevacizumab with palliative intent. In the second line, there is no established standard of care, and inclusion of patients into clinical trials is recommended [71]. The prognosis in patients with metastatic/advanced disease is poor, with a median OS of 5.6–13.3 months after double-agent chemotherapy and up to 16.8 months with the addition of bevacizumab. In second-line settings, treatment with single-agent chemotherapy results in a median OS of <10 months, and there is no clear evidence that it improves survival compared to best supportive care [72,73].

MSI is uncommon in cervical cancers; approximately 3% of cases are MSI-H cases [74]. However, the proportion of TMB-H cases is higher at approximately 15%, and TMB-H should correlate with ICI therapy response, as TMB correlates with neoantigen load in cervical carcinoma [31,38]. Nevertheless, PD-L1 is the most widely studied marker of response. PD-L1 expression is commonly upregulated in cervical cancer, with a prevalence of 55–85% and 64% in squamous cell and adenocarcinoma histology, respectively [75]. However, the numbers depend on the definition of positivity. For example, in the KEYNOTE-158 trial, 83% of patients had a CPS of >1, and in a trial of nivolumab, 75% of patients had a TPS of >1%. In both studies, ICI treatment was overwhelmingly more effective in the PD-L1-positive group [76,77]. 

The use of ICIs in cervical cancer has shown some promise. In the cervical cancer cohort of the KEYNOTE-158 trial, the ORR and DCR were 14.6% and 32.9%, respectively, with no responses in PD-L1-negative tumors [76]. The CHECKMATE-358 trial of nivolumab in 19 cervical cancer patients resulted in an ORR of 26.3% and a DCR of 68.4%. No patient stratification based on established ICI response biomarkers was made; however, patients in the first-line setting had better outcomes than patients after one or more prior systemic therapies [78]. The EMPOWER-Cervical 1 trial on 608 patients with metastatic/advanced cervical cancer demonstrated that cemiplimab as second- and third-line therapy resulted in improved survival with a median OS of 12 months, compared to 8.5 months in patients receiving palliative single-agent chemotherapy. PD-L1 expression correlated with higher efficacy of cemiplimab [79,80,81].

The observed efficacy of ICIs in second-line settings encouraged further investigation in first-line settings. The KEYNOTE-826 trial randomized patients with metastatic/advanced cervical cancer without prior chemotherapy to combined platinum-based doublet chemotherapy with bevacizumab with or without the addition of pembrolizumab. The results indicated improved PFS and OS in the group that received pembrolizumab, with a 24-month OS of 53% in the PD-L1-positive pembrolizumab group vs. 41.7% in the PD-L1-positive placebo group. The results in the all-comer population were only slightly worse (with a 24-month OS of 50.4% and 40.4%); however, almost 90% of the patients expressed PD-L1 [82]. We are still waiting for the results of the BEATcc trial, which combines the ICI atezolizumab with chemotherapy and bevacizumab in a first-line metastatic/advanced setting [83]. The use of ICIs in cervical cancer is increasingly growing. The results of the CALLA and ENGOT-CX11/KEYNOTE-A18 trials, which are evaluating the addition of ICIs to standard treatment (chemoradiotherapy) in locally advanced cervical cancer with positive pelvic or para-aortic lymph nodes, are still pending [84]. 

As preclinical data have indicated the benefits of combined inhibition of more than one co-inhibitory receptor, clinical studies on several tumor types, including cervical tumors, have been performed [85,86]. The combined use of the anti-CTLA4 antibody ipilimumamb and PD-1 antibody nivolumab in recurrent cervical cancer resulted in an ORR of 46%, PFS of 8.5 months, and OS of 25.4 months. This depended on the regimen of administration and whether patients were previously treated with systemic therapy (part of CHECKMATE-358). Patients were not stratified according to PD-1 expression or any other ICI response biomarkers [86]. Similarly, the combination of the PD-1 inhibitor balstilimab and CTLA-4 inhibitor zalifrelimab resulted in an ORR of 22% in the whole population of 155 patients who previously received chemotherapy and an ORR of 27% and 11% in the PD-L1-positive and PD-L1-negative cohort, respectively (RaPiDS trial) [87,88].

As of November 2021, pembrolizumab is FDA-approved as an add-on to the standard-of-care treatment of recurrent or metastatic PD-L1-positive cervical cancer in first-line settings. Additionally, it is approved for PD-L1- or TMB-H-positive cervical cancer in second-line settings [36,89,90]. Conversely, EMA has not yet approved any ICI for the treatment of cervical cancer. Additionally, ESMO guidelines do not mention ICIs in the treatment of cervical cancer. Nevertheless, NCCN guidelines recommend adding pembrolizumab to platinum-based doublet and bevacizumab in first-line systemic therapy if the tumor is PD-L1-positive. Additionally, in second-line systemic therapy, nivolumab can be considered for PD-L1-positive tumors and pembrolizumab for PD-L1-positive, MSI-H/dMMR, and TMB-H tumors [66,91].

### 5.3. Ovarian Cancer

Ovarian cancer is the second most common gynecological malignancy and the seventh most common cancer in women in developed countries, with an estimated 313,000 new cases worldwide and 66,000 in Europe in 2020 [68,92]. As it is most commonly diagnosed in advanced stages (i.e., FIGO stage III and IV), it is the 14th most common cause of cancer-related death worldwide, with 207,000 deaths worldwide and 44,000 in Europe [68]. In ovarian cancer, 90% of tumors are of epithelial origin, of which 70–80% are high-grade serous carcinoma, whereas the others are low-grade serous, mucinous, endometrioid, and clear cell carcinomas [93]. The disease is usually treated by surgical staging and cytoreduction followed by adjuvant systemic therapy. An alternative option for advanced disease is neoadjuvant systemic therapy followed by interval surgical cytoreduction, if feasible. After surgical cytoreduction, systemic therapy is platinum-based doublet (cisplatin/carboplatin and paclitaxel) with the addition of bevacizumab. Patients diagnosed at an early stage have a moderate prognosis: a 5-year recurrence-free survival of 80%. However, 80–90% of patients diagnosed at advanced stages (i.e., FIGO III and IV) eventually relapse. Treatment and prognosis in this setting is determined by the timing of relapse according to discontinuation of systemic treatment, i.e., the platinum-free interval, and stratifies patients with a platinum-free interval of >6 months as either platinum-sensitive or platinum-resistant. A PFS of ≤12 months and an OS of ≤29 months is expected in platinum-sensitive patients. Conversely, a PFS of <6 months has been reported in platinum-resistant patients. Recently, durable responses were observed with the use of PARP inhibitors, namely, olaparib, niraparib, rucaparib, and veliparib. The benefit is the largest in tumors harboring breast cancer gene (BRCA) mutations in patients with homologous recombination deficiency (HRD). For example, maintenance olaparib in patients with recurrence responding to platinum-based chemotherapy resulted in a median PFS of 19.1 months vs. 5.5 months in the placebo group [92,93,94]. 

Already in 2003, it was shown that TILs are present in ovarian cancer tumor tissue in around 50% of cases, which also have higher survival [95]. PD-L1 is also of interest in ovarian cancer. In one study that included all histotypes and grades, tumors were observed with the following characteristics: TPS > 1% (77%), TPS > 10% (11%), CPS > 1 (90%), and CPS > 10 (22%). The results did not significantly differ by histology; however, high-grade serous carcinoma exhibited a trend of having higher TPS and CPS values [96]. Nevertheless, in clinical trials, the responses to ICI therapy were generally low, regardless of biomarker status. In the KEYNOTE-100 trial, the groups with CPS > 1 and CPS > 10 had an ORR of 4.1% and 10%, respectively, indicating the significance of predicting a response based on PD-L1 expression is questionable [97]. Compared to other neoplasms, dMMR is generally rare in ovarian cancer; however, it is highly dependent on specific histotype, varying from 0% of cases in low-grade serous carcinoma to 8% in high-grade serous carcinoma to 21% in ovarian endometrioid cancer. As in endometrial cancer, in patients with Lynch syndrome, there is a higher probability of dMMR in ovarian cancer as well [98,99]. Among gynecological cancers, the prevalence of TMB-H is lowest in ovarian cancer (<5%), with a median number of mutations/Mb of 2–3 [26,100]. Furthermore, in ovarian cancer, TMB is probably not predictive of the ICI response, as it does not correlate with TIL infiltration [38,101].

Ovarian cancers with HRD are associated with higher TMB, neoantigen load, and PD-1 and PD-L1 expression. HRD might correlate with a good response to ICI therapy, as was indicated in a pan-cancer analysis of BRCA-mutated tumors [102,103,104,105]. Ovarian cancer associated with the germline mutated BRCA gene represents approximately 15% of high-grade serous carcinoma cases. However, HRD can be caused by other mechanisms, e.g., somatic (tumor) mutations in BRCA and other genes or BRCA promoter methylation. Hence, the overall prevalence of HRD in ovarian cancer is approximately 50% [106]. There are several tests for HRD in addition to the germline BRCA mutation. The tumor BRCA mutation test on tumor tissue detects and does not discriminate between the germline BRCA and somatic BRCA mutation, which arises de novo in tumor tissue. These tests exhibit good clinical validity. A significant problem with the test for BRCA mutations is the interpretation of variants of unknown significance. Mutations in other genes involved in homologous recombination (HR) and HR gene promotor methylation tests are currently not deemed entirely valid clinically. Genomic scar assays detect genomic damage produced by HRD, e.g., large-scale transitions, loss of heterozygosity, and the number of sub-chromosomal regions with telomeric allelic imbalance. Commercial tests are available, including myChoice by Myriad (for large-scale transitions, loss of heterozygosity, and telomeric allelic imbalance) and FoundationFocus CDxBRCA (for loss of heterozygosity). Whole-genome sequencing by NGS to search for HRD-specific mutational signatures is utilized by HRDetect. Currently, genomic scar assays are deemed clinically valid, whereas mutation signature tests are less so. Phenotypic detection of HRD has an advantage over other tests in that it provides the current status of HRD in tumors. This is important, as reversal mutations in BRCA have been described, with which tumor cells regain their HR mechanism, thus eliminating HRD. However, there is not yet enough evidence to ascertain the clinical validity of these assays [107].

Based on the biological features of ovarian cancer highlighted above, a rationale for an ICI response in ovarian cancer definitely exists. The first clinical studies of ICIs in ovarian cancer were phase 1 with a single agent (nivolumab, pembrolizumab, avelumab, or atezolizumab) and mostly in platinum-resistant patients. Their results were not promising, with ORRs of 6–22% [97,108,109,110,111,112]. A phase 3 trial of nivolumab vs. single-agent chemotherapy in platinum-resistant patients yielded poor results; the OS was similar in both groups, and the median PFS was better in the chemotherapy group (3.8 vs. 2 months) [113].

Trials of ICIs in combination with other agents are described below. JAVELIN Ovarian 200, a phase 3 trial on platinum-resistant patients, resulted in a non-significant increase in the PFS of patients treated with avelumab plus single-agent chemotherapy vs. only single-agent chemotherapy (3.7 vs. 3.5 months) [114]. Interestingly, JAVELIN Ovarian 100, a phase 3 trial on platinum-sensitive patients, did not show any benefit of adding avelumab to standard-of-care platinum-based chemotherapy and indicated a better outcome with only chemotherapy. The median PFS was 16.8–18.1 months in the combination group (depending on the regime), and median progression-free survival (mPFS) was not achieved in the chemotherapy-only group. In a subgroup analysis of this trial, stratifying patients by PD-L1 status and BRCA mutation did not reveal differences [115,116]. Other, more complex combination strategies were explored as well. For example, in a phase 2 trial on platinum-sensitive and platinum-resistant patients (n = 40), pembrolizumab was combined with single-agent chemotherapy. In this trial, oral cyclophosphamide was metronomically administered, which means that frequent (daily) low doses were used. The ORR was 47.5%, with a durable response (>12 months) in 25% of patients [117].

Another strategy is combination with antiangiogenic agents, as was used in the IMagyno-050 trial. This phase 3 trial enrolled 1300 patients at diagnosis of ovarian cancer who received platinum-based double-agent chemotherapy, bevacizumab with or without atezolizumab. The median PFS in the group receiving atezolizumab was 19.5 months, compared to 18.4 months in the group receiving only bevacizumab; however, the difference was not significant. Nevertheless, the difference between the groups was significant (mPFS 20.8 vs. 18.5 months) in the PD-L1-positive population, but of dubious clinical meaning, as there was no difference in OS [118]. In LEAP-005, a phase 2 trial on 31 heavily pretreated ovarian cancer patients administered the multi-tyrosine inhibitor lenvatinib in combination with pembrolizumab. The ORR was 32% [119].

The strategy of combining PARP inhibitors with ICIs was explored in two trials. TOPACIO, a phase 1 and 2 trial on 62 platinum-resistant patients administered niraparib in combination with pembrolizumab, demonstrated an ORR of 18%, and, of note, some responses were durable for up to 20 months. Exploratory subgroup analysis using PD-L1 expression, tumor BRCA mutation, or HRD status did not show any significant correlation with efficacy [120]. MEDIOLA, a phase 2 study on 32 patients with non-germline BRCA-mutated, platinum-sensitive disease, demonstrated an ORR of 87% in patients treated with triple therapy: olaparib, durvalumab, and bevacizumab [121,122]. The NRG GY003 trial combined multiple ICIs and included 100 patients with recurrent ovarian cancer after 1–3 prior systemic treatments with platinum-free intervals of <12 months. Patients were randomized to either nivolumab plus pembrolizumab or nivolumab alone. Combination therapy was significantly more effective, with a PFS of 3.9 months vs. 2 months after single-agent therapy. However, as the combination therapy caused more AEs, the clinical meaning of this prolongation in survival is questionable [123]. Multiple trials of the combination of ICIs, anti-angiogenic agents, PARP inhibitors, and chemotherapy in several settings (adjuvant, maintenance, recurrent platinum-sensitive, or platinum-resistant) are currently taking place [124,125,126,127,128].

Although the rationale for immunotherapy in ovarian cancer is clear, this therapy has not yet lived up to its promise in clinical trials. Additionally, conventional (MSI/MMR, PD-L1 expression, and TMB) and ovarian-cancer-specific (BRCA mutation and HRD) biomarkers have not effectively distinguished between patients who benefit most from ICIs. This may be due to the overall low response rates in ovarian cancer. In the future, combinations of ICIs with other antineoplastic agents may provide better results. The development of immunotherapies that target other co-inhibitory receptors and their ligands or stimulatory receptors on immune cells might yield novel strategies for treating this disease. Concurrently, the discovery of new biomarkers of response to clinically established or still emerging immunotherapies will probably be needed to stratify patients, first for clinical trials and subsequently for routine practice.

As of November 2021, ESMO guidelines do not mention the use of ICIs in epithelial or non-epithelial ovarian cancer. NCCN guidelines deem immunotherapy useful for treating both recurrent platinum-resistant and platinum-sensitive epithelial ovarian cancer. Specifically, pembrolizumab for MSI-H/dMMR or TMB-H tumors and dostarlimab for MSI-H/dMMR tumors in patients with no satisfactory alternative treatment options is mentioned [66,129].

### 5.4. Vulvar and Vaginal Cancer

With an incidence of approximately 15,000 new cases estimated in Europe in 2020, vulvar and vaginal cancer is uncommon, accounting for only 2–5% of gynecological malignancies [68,130]. Histologically, it is mainly squamous cell carcinoma. In postmenopausal women, it typically arises from a preneoplastic lesion termed differentiated VIN and associated with lichen sclerosus. In younger patients, it typically occurs from preneoplastic lesions associated with human papillomavirus (HPV) infection, as in cervical cancer [130]. The disease is usually diagnosed at an early stage, and surgery and (chemo)radiotherapy are the mainstay of treatment [130,131]. For metastatic disease, palliative systemic therapy should be considered, as no optimal regimen has been established to date and limited data demonstrates only poor responses [131].

In clinical studies to date, an extremely small number of vulvar cancer patients were treated with ICIs. Generally, the responses were poor. The KEYNOTE-028 trial comprised 18 patients with PD-L1-positive vulvar squamous cell carcinoma treated with pembrolizumab and demonstrated an ORR of <10% and a median OS of 3.8 months, with only one patient achieving a partial response. PD-L1 expression positivity as an inclusion criterion was defined as ≥1% modified proportion score or interface pattern by IHC [132]. In this basket trial, including >20 different cancers, PD-L1, TMB, and gene expression profiles were studied as biomarkers. The biomarker status was not assessed in all patients. PD-L1 expression was reported as the CPS by IHC (a different method compared to the inclusion criterion). The TMB status was assessed by whole-exome sequencing. The T-cell-inflamed gene expression profile was assessed by a method previously described [24]. The biomarker status of a specific cancer cohort was not reported [25]. A study of nivolumab in five vulvar cancer patients examined the PD-L1 status. All patients assessed were positive. One partial response and three patients with stable disease were observed; the ORR was 20%, and the DCR was 80% [78]. A clinical study of pembrolizumab combined with chemoradiation in unresectable locally advanced or metastatic vulvar cancer is currently underway [133]. Additionally, case reports on three PD-L1-positive patients with recurrent or metastatic disease demonstrated that two patients had a complete or partial response, one of them even after discontinuing therapy, and one patient progressed during therapy [134,135].

As of November 2021, ESMO guidelines for vulvar cancer do not exist. The European Society of Gynecological Oncology guidelines do not recommend any specific systemic therapy regimen for vulvar cancer and do not mention ICIs [66,131]. NCCN guidelines deem ICIs useful, specifically pembrolizumab for MSI-H/dMMR, PD-L1-positive, or TMB-H tumors in second-line settings and nivolumab for HPV-related advanced, recurrent, or metastatic vulvar cancer [136].

Selected studies of ICI monotherapy or combination therapy in gynecological cancers, with emphasis on biomarker-based subgroup responses, are shown in Table 2.

## 6. Discussion and Conclusions

ICI monotherapy is, to date, only modestly effective for gynecological cancers. The exception is MSI-H/dMMR and, to some extent, MMRp endometrial cancer, which carries the best prognosis among gynecological cancers overall, even with standard-of-care therapies. Smaller studies indicate some response to ICI therapy when combined with chemotherapy, anti-angiogenic agents (e.g., bevacizumab), or other targeted antineoplastic agents. There is still a wide variety of clinical scenarios with numerous patients having unmet clinical needs, especially recurrent platinum-resistant ovarian cancer and advanced or metastatic cervical carcinoma. Nevertheless, this is an extremely developing field with several studies with innovative strategies awaiting data. In the future, we expect that combinations of ICIs with therapies that may make immune cold tumors more immunoreactive could yield positive results. Furthermore, novel immune pathways and targets are constantly being explored, and new agents are being developed. Examples of other immune inhibitory receptors with targeting agents in clinical development are: LAG3, TIM3, VISTA, TIGIT. Another approach is to target T-cell co-stimulatory receptors with agonist antibodies. Targeting those receptors can be described as “stepping on gas” in contrast to “stepping off the brake” with inhibitory receptor inhibition. Examples of costimulatory receptors being targeted with agents that are currently in clinical development are: 4–1BB, CD40, CD27, GITR, ICOS. Another logical step is combining several inhibition strategies, combining stimulatory strategies, or both. However, as expected, combination strategies are associated with profoundly higher incidence and severity of adverse effects [10,14,138]. 

There are numerous benefits to having a marker or tool that predicts the response to ICI therapy. When faced with a low probability of an ICI treatment response, the treating physician can choose other, potentially more effective treatments, thus sparing the patient from potentially severe AEs and avoiding the high costs associated with cancer immunotherapy. Currently, ICI response prediction based on biomarker status faces several obstacles, including non-standardized assays, various cut-offs, non-comprehensive reporting of biomarker status, and dependence of the utility of a biomarker on specific histotypes and clinical setting (presentation/recurrence). This is explained by the fact that clinical research of ICIs has, to some extent, surpassed our understanding of the basic mechanisms that drive responses to treatment. Additionally, due to the small overall number of gynecological cancer patients treated in ICI studies, subgroup analyses of responses based on biomarker status are frequently underpowered. To date, our understanding of this topic has grown, and hopefully, clinical researchers will design studies incorporating robust detection of several biomarkers with standardized methodologies and comprehensive reporting on the detection methods and patient’s biomarker status. Some progress in the field has already been made with attempts to standardize biomarker detection [21,23,35]. 

Our overall impression is that not only one but several biomarkers combined with patient and tumor characteristics will guide clinical decision-making in the future. Response prediction models based on biomarker status and patient/tumor characteristics have already been developed and retrospectively tested; however, they still need validation with prospective clinical studies [139]. 

Biomarker status can change during the course of the disease, and this problem has, to date, only been resolved with repeat biopsies, an invasive procedure. Blood-based biomarkers could solve both of these problems. Concurrently with the clinical development of agents targeting other immune inhibitory or stimulatory pathways, biomarker development should take place, which is already happening to some extent. This is especially important in the light of the higher incidence and severity of ICI treatment-related adverse effects, as described above.

## Figures and Tables

**Figure 1 cancers-14-00631-f001:**
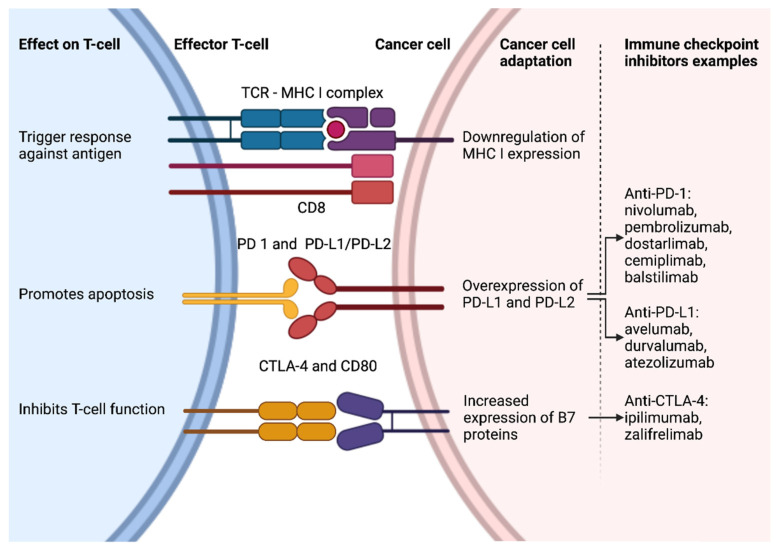
Mechanism of action and development of ICIs. Created with ©BioRender.

**Table 1 cancers-14-00631-t001:** Overview of selected ICI treatment response prediction biomarkers.

Biomarker	Rationale	Method of Detection	Use in Gynecological Cancer	Discriminating Result
Tumor-infiltrating lymphocytes (TILs)	Presence of TILs is indicator of intrinsic immune response	Histopathology of TT, semi-quantitatively, spatial pattern (1. between tumor cells, 2. in stroma, 3. absence)	No	1. Immune-hot tumors 2. Immune-excluded tumors 3. Immune-deserted tumors
PD-1/PD-L1 expression	PD-L1 on tumor cells indicate activated immune escape mechanisms, PD-1 on TILs indicates T-cell exhaustion	Histopathology of TT -> IHC staining, semi-quantitatively; 1. Expression on tumor cells—tumor proportion score (TPS) 2. Expression in TT—combined proportion score (CPS) 3. Pattern of IHC staining;	1. Uterine cervical cancer 2. Ovarian cancer 3. Vulvar cancer	Score expressed as number or percent; cut-offs varied from 1%-50% in studies, depending on tumor histotype
TME gene expression profiles	Immune-related gene expression is a marker of immune response	mRNA isolation from TT -> gene expression quantification -> model-based score	No	Not clearly defined
Tumor neoantigen load	Tumor neoantigens initiate immune response	DNA isolation from TT -> NGS -> in silico neoantigen prediction	No	Not clearly defined
Tumor mutational burden (TMB)	High TMB = hypermutated genome -> high tumor neoantigen load	DNA isolation from TT ->NGS(WES)/targeted assays (FoundationOne, MSK-IMPACT)	Potentially all (tissue FDA approval of pembrolizumab)	TMB-high -> 10–20 mut/Mb (varies)
Microsatellite instability (MSI), mismatch repair deficiency (MMR)	Defective DNA repair mechanism -> hypermutated genome -> high tumor neoantigen load	1. MMR—histopathology of TT with IHC for 4 proteins involved in MMR 2. MSI—TT DNA isolation -> PCR with 5 probes	1. Endometrial cancer 2. Ovarian cancer	1. Non-expression in ≥ 1 protein -> MMRd (otherwise pMMR) 2. Altered repeat length in ≥ probes—MSI-H (otherwise MSS)
Homologous recombination deficiency (HRD)	Defective DNA repair mechanism -> hypermutated genome -> high tumor neoantigen load	1. Germline BRCA mutation (genetic testing of normal tissue) 2. Tumor BRCA mutation (genetic testing of TT) 3. Genomic scar assay (Myriad myChoice, FoundationFocusCDxBRCA) 4. Other, less clinically validated assays	Ovarian cancer	1. gBRCA mut/wt 2. tBRCA mut/wt 3. HRD +/−

Mut—mutation/mutated; wt—wild type; MSS—microsatellite stable; Mb—mega base pair (10^6^ base pairs).

**Table 2 cancers-14-00631-t002:** Selected studies of ICI monotherapy or combination therapy in gynecological cancers, with emphasis on biomarker-based subgroup responses. In most studies, not all patients were evaluated regarding biomarker status, notable in cases where the numbers do not add up to total number of patients. In several studies, subgroup analyses based on biomarker status reported outcomes should be interpreted cautiously, as they have not been powered enough.

Trial	Phase	Intervention	Study Population and Biomarker Status (Number of Patients)			Outcomes	
**Endometrial Cancer—ICI Monotherapy**							
KEYNOTE—158 [37]	2	Pembro	A/R/M disease, ≥1 PST	MSI-H/dMMR (49)		ORR 57%, mPFS 25.7 mo	
				TMB-H (15) ^a^		ORR 46%	
				Non-TMB-H (67) ^a^		ORR 6%	
GARNET [54]	1	Dostarlimab	A/R/M disease, ≥1 PST	MSI-H/dMMR (103)		ORR 46% (CR in 10.7%), DCR 59%	
				MSS/MMRp (156)	TMB-H ^a^ (141)	ORR 13%	ORR 45.5%
					Non TMB-H ^a^ (13)	DCR 35%	ORR 12.1%
**Endometrial Cancer—ICI Combined with Other Agents**							
KEYNOTE-775 [60]	3	Pembro + Lenvatinib vs. Chemo	A/R/M disease, ≥ 1 PST	All patients (827)		ORR 31.9% vs. 14.7%; mPFS 7.2 vs. 3.8 mo; mOS 18.3 vs. 11.4 mo	
				MSI-H/dMMR (130)		ORR 40% vs. 12.3%; mPFS 10.7 vs. 3.7 mo; mOS NR vs. 8.6 mo	
				MSS/MMRp (697)		ORR 30.3% vs. 15.1%; mPFS 6.6 vs. 3.8 mo; mOS 17.4 vs. 12.0 mo	
KEYNOTE-158 [76]	2	Pembro	A/R/M disease, 78% pts received ≥1 PSTs	All patients (98)		ORR 12.2%, DCR 30.6%	
				PD-L1+ ^b^ (82)		ORR 14.6%, DCR 32.9%	
				PD-L1– ^b^ (15)		ORR 0%, DCR 20%	
CHECKMATE-358 [78]	1/2	Nivo	A/R/M disease, ≥1 PST	All patients (24), of those 10 PD-L1+, 6 PD-L1−, 3 NA ^c^		ORR 26.3%, DCR 68.4%, mPFS 5.1 mo	
EMPOWER-Cervical 1 [79]	3	Cemiplimab vs. Chemo	Recurrent/metastatic ≥1 prior syst. Th.	All patients (608)		mOS 12 v 8.5 mo	
				PD-L1+ ^d^ (162)		mOS ~ 14.5 vs. 9 mo	
				PD-L1– ^d^ (92)		mOS ~ 8 vs. 6 mo	
**Uterine Cervical Cancer—ICI Combined with Other Agents**							
KEYNOTE-826 [82]	3	SoC + pembro vs. SoC (Platinum-based doublet + bev)	A/R/M, no PST (first line)	All patients (619)		mPFS 10.4 vs. 8.2; HRPO 0.65	
				PD-L1 CPS 1 (35, 11%)		HRPO 0.94	
				PD-L1 CPS ≥ 1 (548, 88%)		mPFS 10.4 vs. 8.2 mo; HRPO 0.62	
				PD-L1 CPS >10 (158, 51%)		mPFS 10.4 vs. 8.1 mo; HRPO 0.58	
**Uterine Cervical Cancer—ICI Combined with Another ICI**							
CHECKMATE-358 [86]	1/2	Ipi + nivo	A/R/M, 0–2 PST	All patients(91)		ORR 46%, mPFS 8.5 mo, mOS 25.4 mo	
RaPiDS [87]	2	Balstilimab+/−zalifrelimab	A/R/M, ≥1 PST	Combination group (143)	All (143)	ORR 22%	
					PD-L1+ ^b^ (55%)	ORR 27%	
					PD-L1– ^b^ (25%)	ORR 11 %	
					PD-L1 NA (20)	ORR 21%	
**Ovarian Cancer—ICI Monotherapy**							
NINJA [113]	3	Nivo vs. single -agent chemo (GEM or PLD)	Relapsed, platinum resistant	All (316)		mPFS 2 vs. 3.8 mo; mOS 10.1 vs. 12.1 (favours chemo)	
				PD-L1+ (123) ^d^			
				PD-L1− (189) ^d^			
**Ovarian Cancer—ICI Combined with Other Agents**							
JAVELIN Ovarian 200 [114]	3	Ave (188) vs. Ave + PLD (188) vs. PLD (190)	Relapsed, platinum -resistant or refractory (no PST for platinum resistant disease)	All (566)	PD-L1+ ^e^ (288)	ORR 4% vs. 13% vs. 4%; DCR 33% vs. 49% vs. 57%	
					PD-L1– ^e^ (220)		
					TIL+ ^f^ (228)		
					TIL– ^f^ (227)		
JAVELIN Ovarian 100 [115]	3	PDC + maintenance Ave vs. PDC + Ave + maintenance Ave vs. PDC	First line—ACT or NACT	All (998)	PD-L1+ ^e^ (477)	mPFS 16.8 vs. 18.1 vs. NE Mo; HRP 1.43 vs. 1.14 vs. 1 (results favour PDC)	HRP 1.23 vs. 0.98 vs. 1
					PD-L1– ^e^ (326)		HRP 1.02 vs. 1.36 vs. 1
					gBRCA mut. (93)		HRP 1.98 vs. 2.51 vs. 1
					gBRCA wt (854)		HRP 1.32 vs. 1.14 vs. 1
Kunde et.al. [117]	2	Pembro + metronomical CPA + bev	Relapsed, platinum—sensitive (25%) and platinum—resistant (75%)	40 patients	BRCA ^g^ mut 35%	ORR 47.5%, DCR 95%, mPFS 10 mo	/
					BRCA ^g^ wt 57.5%		/
					PD-L1+ ^h^ 47.5%		ORR 52.6%
					PD-L1– ^h^ 42.5%		ORR 35.3%
IMagyno-050 [118]	3	PDC + bev vs. PDC + bev + atezolizumab	First line—ACT or NACT	All patients (1301)		mPFS 18.4 vs. 19.5 (HRP 0.92)	
				PD-L1 < 1% IC ^i^ (517)			
				PD-L1 ≥ 1% IC ^i^ (784)			
				PD-L1 ≥ 5% IC ^i^ (260)			
				PD-L1 >= 1% TC ^i^ (73)			
LEAP-005 [119]	2	Lenvatinib + pembro	Relapsed, 3 PST, (80% platinum resistant/refractory)	All patients (31)		ORR 32%, DCR 74%, mPFS 4.4 mo	
TOPACIO/KEYNOTE-162 [120]	1/2	Niraparib + pembro	Relapsed, platinum sensitive and resistant disease, 1–5 PST; median number of PSTs was 3	All patients (60)		ORR 18%, DCR 65%	
				tBRCA^j^	Mut(11, 18%)	ORR 18%	
					Wt(49, 79%)	ORR 18%	
				PD-L1 ^k^	+(35, 56%)	ORR 21%	
					−(21, 34%	ORR 10%	
				HRD^b^	+(22, 35%)	ORR 14%	
					−(33, 53%	ORR 19%	
MEDIOLA [121,137]	2	Olaparib + durva	Relapsed, platinum sensitive, ≥ 1 PST	gBRCA mut (32) (doublet)		ORR 71.9%, mPFS 11.1 mo	
				sBRCA mut. (32) (doublet)		ORR 31 %, mPFS 5.5 mo	
		Olaparib + durva + bev		sBRCA mut. (31) (triplet)		ORR 77%, mPFS 14.7 mo	
**Ovarian Cancer—ICI Combined with Another ICI**							
NRG-GY003 [123]	2	Nivo vs. Nivo + ipi	Relapsed, 1–3 PST, platinum resistant or platinum sensitive, PFI < 12 mo	All (100)		mPFS 2 vs. 3.9 mo; mOS 22 vs. 28 mo	
		Nivo or Nivo + Ipi (pooled data)		Any PD-L1 in TC ^l^	+(5)	ORR 36%, mPFS 2.5 mo	
					−(26)	ORR 23%, mPFS 4 mo	
				PD-L1 ≥ 1% in IC ^l^	+(20)	ORR 31%, mPFS 4 mo	
					−(11)	ORR 19%, mPFS 2.3 mo	
**Vulvar Cancer**							
KEYNOTE-028 [25]	1 ^b^	Pembro	Advanced, PD-L1 positive ^h^	All (18)		ORR 6%, mPS 3.1 mo, mOS 3.8 mo	
CHECKMATE-358 [78]	1/2	Nivo	Advanced	All (5), of those 4 PD-L1+ ^m^, 1 pt NA		ORR 20%, DCR 80%	

^a^—TMB-H was defined as ≥ 10 mut/Mb as per FoundationOne CDx assay; ^b^—tumors with PD-L1 CPS ≥ 1 and <1 by IHC were regarded as PD-L1 positive or negative, respectively; ^c^—Tumor cell PD-L1 expression was defined as the percentage of tumor cells exhibiting plasma membrane staining at any intensity; ^d^—tumors with PD-L1 TPS ≥ 1 and <1 by IHC were regarded as PD-L1 positive or negative, respectively; ^e^—A sample was considered PD-L1- positive if either at least 1% of assessed tumor cells expressed membranous PD-L1, at least 5% of immune cells within the tumor area expressed PD-L1, or both; ^f^—Tumor infiltrating lymphocytes (CD8+) are associated with prognosis. A sample was considered TIL positive if at least 1% of cells within the tumor area expressed CD8; ^g^—in this study mutation status included germline and somatic mutations; ^h^—PD-L1 positivity was defined by the presence of the interface pattern staining or a modified; proportion score ≥1%; ^i^—in this study PD-L1 status was segregated in more groups, depending on the IHC staining on immune cells (IC) or tumor cells (TC). Results in PD-L1 ≥ 1% TC were encouraging, but the group had small number of patients who largely overlap with PD-L1 ≥ 1% IC population; ^j^—BRCA mutation status was assessed using the Myriad genetics assay; ^k^—HRD status was assessed using the Myriad genetics assay; ^l^—in this study PD-L1 status was determined separately on tumor or immune cells by IHC staining; In contrast to other studies, this also included a category where any staining of tumor cells was regarded as PD-L1 positive; ^m^—PD-L1 + was defined as PD-L1 ≥ 1% TC or PD-L1 CPS > 1. PLD: pegylated liposomal doxorubicin; PST: past systemic therapy; HRD: homologous recombination deficiency; vs.: versus; ACT: adjuvant chemotherapy; NACT: neoadjuvant chemotherapy; PDC: platinum-based doublet chemotherapy; Maint.: maintenance; NE: not estimable; 5%PFS: lower bound of 95% CI for PFS (as mPFS is not estimable in all subgroups); Mo: months; HRP: HR (hazard ratio) for PFS; HRPO: HR (hazard ratio) for PFS or OS; CPA: cyclophosphamideMut.: mutated; Nivo: nivolumab; Ipi: ipilimumab; TC: tumor cells; IC: immune cells; NA: not assessed; Chemo: chemotherapy; Ave: avelumab; Bev: bevacizumab; A/R/M: advanced/recurrent/metastatic; SoC: standard of care; GEM: gemcitabine; gBRCA: germline BRCA status (mutated (mur) /wild-type (wt)); tBRCA: tumor/somatic BRCA status (mutated (mur) /wild-type (wt)).

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
