# Peer review of "Overview of Immune Checkpoint Inhibitors in Gynecological Cancer Treatment"

_cancers, 2022, doi:10.3390/cancers14030631_

Round 1
Reviewer 1 Report
In this review article, the authors have given a comprehensive summary on the immune checkpoint inhibitor (ICI) treatments in gynecological cancer based on 126 papers in this exploding research field in the last decade. They reviewed the basic biology of ICIs and biomarkers for response prediction. The findings from 20 clinical trials were organized and presented in Table 1 so that it is easier for readers to compare the results from different studies. The following are suggestions to improve the article:
- The structure of the article is good, but it can be further improved by adding some subsection titles. The section 3 has only one subsection 3.1, and it is better to add the ” section 3.1 ICI basic biology” before the line 65. This section is about cancer immunotherapy in general. It is better to add some references from people such as James Allison, Tasuku Honjo and Steven Rosenberg who are pioneers in this field.
- The response prediction of ICI treatment in gynecological cancer is an important and challenging problem. However, this article also includes other materials such as the summary of studies and clinical trials, opinions on the ICI treatment plan, immunotherapy and combination therapies, and clinical guidelines etc. not directly related to the response prediction of ICI treatment. The title does not match its content. It is better to change the title to “Overview of ICI treatment in gynecological cancer”.
- The microbiome is a key biomarker which is missing in the section 4.
- It will be useful for readers if a table of biomarkers is added.
- Some acronyms were not defined: on P.7, line 294, ICB; and on P.14, Table1, Wt, wt, PST, PD-L1d, and HRDd.
- The authors need to double check the layout of Table 1 and make sure all acronyms are defined and the superscripts are at the right position.
- On P.8, line 361, the word “and” is not needed.
Author Response
Response to reviewers’ comments
We thank the reviewer for her/his valuable insights and constructive comments that have contributed to a better quality of our manuscript. Below are our responses in italics:
Responses to reviewer 1
In this review article, the authors have given a comprehensive summary on the immune checkpoint inhibitor (ICI) treatments in gynecological cancer based on 126 papers in this exploding research field in the last decade. They reviewed the basic biology of ICIs and biomarkers for response prediction. The findings from 20 clinical trials were organized and presented in Table 1 so that it is easier for readers to compare the results from different studies. The following are suggestions to improve the article:
- The structure of the article is good, but it can be further improved by adding some subsection titles. The section 3 has only one subsection 3.1, and it is better to add the ” section 3.1 ICI basic biology” before the line 65. This section is about cancer immunotherapy in general. It is better to add some references from people such as James Allison, Tasuku Honjo and Steven Rosenberg who are pioneers in this field.
Thank you for thoughtful suggestion. First part in the section 3 describes basic tumor immunobiology, hence we named it “3.1. Overview of tumor immunobiology” Second part is about ICIs, currently used in clinical practice, hence we named it “3.2. Overview of immune checkpoint inhibitors”. We added relevant citations from the pioneering authors mentioned.
- The response prediction of ICI treatment in gynecological cancer is an important and challenging problem. However, this article also includes other materials such as the summary of studies and clinical trials, opinions on the ICI treatment plan, immunotherapy and combination therapies, and clinical guidelines etc. not directly related to the response prediction of ICI treatment. The title does not match its content. It is better to change the title to “Overview of ICI treatment in gynecological cancer”.
We are grateful for the suggestion and agree with it. In line with it we changed the article title.
- The microbiome is a key biomarker which is missing in the section 4.
Thank you for pointing out this exciting novel discovery in the field. Microbiome as a biomarker as well as potential therapeutic target is now reviewed in the added section 4.6.
- It will be useful for readers if a table of biomarkers is added.
We agree with the reviewer, hence we added the table of biomarkers reviewed to the paper.
- Some acronyms were not defined: on P.7, line 294, ICB; and on P.14, Table1, Wt, wt, PST, PD-L1d, and HRDd.
- The authors need to double check the layout of Table 1 and make sure all acronyms are defined and the superscripts are at the right position.
Thank you for pointing out mistakes in the table. We thoroughly reviewed the table once again, corrected typing mistakes, defined all abbreviations and sorted superscripts by the order of appearance in the table.
- On P.8, line 361, the word “and” is not needed.
We corrected the typing mistake and others we found after thoroughly reading the paper again.
Reviewer 2 Report
Summary
Tumors develop in a complex ecology. Through direct and indirect cell-cell interactions, malignant and non-malignant stromal cells communicate and shape the tumor microenvironment (TME) to facilitate growth, immune evasion and dissemination. Traditionally, targeted therapies were tailored to interfere with cancer-intrinsic drivers of growth. However, the recent success of immune checkpoint blockade therapies (ICB) demonstrated the vast potential of harnessing the immune system to eliminate malignant cells in local and disseminated disease. Nevertheless, the efficacy of current ICB therapies is limited (Yarchoan et al., 2017), as they rely on high mutation burden (neoantigens) that are found in only a subset of cancers, and require robust activation of the adaptive immune system which is heavily suppressed by tumor-infiltrating myeloid cells. These limitations narrow the spectrum of cancer patients that benefit from these therapies and represent a major clinical herdle.
Pirš et. al. comprehensively reviewed the current state of ICB therapies in gynecological cancers, spanning recently concluded and on-going clinical trials of different ICB regiments The authors listed several biomarkers, including expression of mismatch repair proteins and PD1/PD-L1, the field uses to evaluate patients enrollment for ICB therapies based on response rates. However, these biomarkers fail to predict the efficacies in gynecological cancers including Ovarian cancer, indicating more mechanisms are involved.
The authors provided a detailed snapshot of the current success rates in gynecological cancers and different clinical trials. This provides the readers a clear understanding of the current unmet clinical challenges in these cancers, however the perspective of the authors on how to tackle these challenges and underlying mechanisms is missing. The readers would benefit from presentation of a well-known problem and suggested solutIons from a clinical perspective. Overall, the manuscript is well written, organized and data is clear. Therefore, I recommend for publication pending several modifications listed below.
Points to address:
- When the authors describe immune escape mechanisms via MHC-I downregulation, and low mutation burden, please cite the relevant papers that were the first to address. For example the mutations in b2M gene in recurrent melanoma patients: “Mutations Associated with Acquired Resistance to PD-1 Blockade in Melanoma” (Snyder et al., 2014; Zaretsky et al., 2016).
- Immune suppression and expression of PD-L1 in the TME is mediated by myeloid-derived suppressor cells and tumor-associated macrophages (Cassetta and Pollard, 2018; DeNardo and Ruffell, 2019). Please address in more detail the immune suppression aspect in TME as potential mechanisms that limit efficacies of ICB therapies.
- The authors stated “AEs are more common after use of anti-CTLA4 vs anti-PD-1/PD-L1 ICIs” (line 122-123). Please address the bias toward CTLA4 treatment (vs PD1/PDL1) given the fundamental difference in mechanism of actions. CTLA4 inhibition allows persistent activation of immune stimulation from antigen presenting cells (CD28-CD86/80).
- Typo in line 267. Please correct.
- Please address other immune checkpoint pathways beyond PD1/PD-L1, particularly given the limited success of PD1/PD-L1 expression to predict response.
Other pathways to consider include TIM3-Galectin9, CD155-TIGIT, ICOS-ICOSL, GITR-GITRL.
References
Cassetta, L., and Pollard, J.W. (2018). Targeting macrophages: therapeutic approaches in cancer. Nat. Rev. Drug Discov.
DeNardo, D.G., and Ruffell, B. (2019). Macrophages as regulators of tumour immunity and immunotherapy. Nature Reviews Immunology 19, 369–382.
Snyder, A., Makarov, V., Merghoub, T., Yuan, J., Zaretsky, J.M., Desrichard, A., Walsh, L.A., Postow, M.A., Wong, P., Ho, T.S., et al. (2014). Genetic basis for clinical response to CTLA-4 blockade in melanoma. N. Engl. J. Med. 371, 2189–2199.
Yarchoan, M., Hopkins, A., and Jaffee, E.M. (2017). Tumor Mutational Burden and Response Rate to PD-1 Inhibition. N. Engl. J. Med. 377, 2500–2501.
Zaretsky, J.M., Garcia-Diaz, A., Shin, D.S., Escuin-Ordinas, H., Hugo, W., Hu-Lieskovan, S., Torrejon, D.Y., Abril-Rodriguez, G., Sandoval, S., Barthly, L., et al. (2016). Mutations Associated with Acquired Resistance to PD-1 Blockade in Melanoma. N. Engl. J. Med. 375, 819–829.
Author Response
Response to reviewers’ comments
We thank the reviewer for her/his valuable insights and constructive comments that have contributed to a better quality of our manuscript. Below are our responses in italics:
Responses to reviewer 2
Tumors develop in a complex ecology. Through direct and indirect cell-cell interactions, malignant and non-malignant stromal cells communicate and shape the tumor microenvironment (TME) to facilitate growth, immune evasion and dissemination. Traditionally, targeted therapies were tailored to interfere with cancer-intrinsic drivers of growth. However, the recent success of immune checkpoint blockade therapies (ICB) demonstrated the vast potential of harnessing the immune system to eliminate malignant cells in local and disseminated disease. Nevertheless, the efficacy of current ICB therapies is limited (Yarchoan et al., 2017), as they rely on high mutation burden (neoantigens) that are found in only a subset of cancers, and require robust activation of the adaptive immune system which is heavily suppressed by tumor-infiltrating myeloid cells. These limitations narrow the spectrum of cancer patients that benefit from these therapies and represent a major clinical herdle.
Pirš et. al. comprehensively reviewed the current state of ICB therapies in gynecological cancers, spanning recently concluded and on-going clinical trials of different ICB regiments The authors listed several biomarkers, including expression of mismatch repair proteins and PD1/PD-L1, the field uses to evaluate patients enrollment for ICB therapies based on response rates. However, these biomarkers fail to predict the efficacies in gynecological cancers including Ovarian cancer, indicating more mechanisms are involved.
The authors provided a detailed snapshot of the current success rates in gynecological cancers and different clinical trials. This provides the readers a clear understanding of the current unmet clinical challenges in these cancers, however the perspective of the authors on how to tackle these challenges and underlying mechanisms is missing. The readers would benefit from presentation of a well-known problem and suggested solutIons from a clinical perspective. Overall, the manuscript is well written, organized and data is clear. Therefore, I recommend for publication pending several modifications listed below.
Points to address:
- When the authors describe immune escape mechanisms via MHC-I downregulation, and low mutation burden, please cite the relevant papers that were the first to address. For example the mutations in b2M gene in recurrent melanoma patients: “Mutations Associated with Acquired Resistance to PD-1 Blockade in Melanoma” (Snyder et al., 2014; Zaretsky et al., 2016).
We thank the reviewer for pointing out this mechanisms of tumor immune escape. We expanded section 3.1 with information on b2M mutation and relevant citation in section 3.1 and stressed importance of neoantigen expression pattern with relevant citation in section 4.1.
- Immune suppression and expression of PD-L1 in the TME is mediated by myeloid-derived suppressor cells and tumor-associated macrophages (Cassetta and Pollard, 2018; DeNardo and Ruffell, 2019). Please address in more detail the immune suppression aspect in TME as potential mechanisms that limit efficacies of ICB therapies.
Thank you for mentioning this aspect of TME immunesuppresive properties. Information about TAM and MDSC with relevant citations was added to section 3.1
3.The authors stated “AEs are more common after use of anti-CTLA4 vs anti-PD-1/PD-L1 ICIs” (line 122-123). Please address the bias toward CTLA4 treatment (vs PD1/PDL1) given the fundamental difference in mechanism of actions. CTLA4 inhibition allows persistent activation of immune stimulation from antigen presenting cells (CD28-CD86/80).
In section 3.1 we explained difference in anti-CTLA4 vs. anti-PD-1/L1 mechanism of action explaining different AE profile.
- Typo in line 267. Please correct.
We corrected the typing mistake and others we found after thoroughly reading the paper again.
- Please address other immune checkpoint pathways beyond PD1/PD-L1, particularly given the limited success of PD1/PD-L1 expression to predict response.
Other pathways to consider include TIM3-Galectin9, CD155-TIGIT, ICOS-ICOSL, GITR-GITRL.
Thank you for reminding us about novel developments in the field. We expanded discussion and conclusions section (section 6) with information about other immune inhibitory pathways as well activation pathways and explained how targeting them differs. We added relevant citations in this section supporting the text.